# Beyond Henssge’s Formula: Using Regression Trees and a Support Vector Machine for Time of Death Estimation in Forensic Medicine

**DOI:** 10.3390/diagnostics13071260

**Published:** 2023-03-27

**Authors:** Lívia Mária Dani, Dénes Tóth, Andrew B. Frigyik, Zsolt Kozma

**Affiliations:** 1Department of Forensic Medicine, University of Pécs Medical School, Szigeti út 12, H-7624 Pécs, Hungary; 2Bánki Donát Faculty of Mechanical and Safety Engineering, Óbuda University, Népszínház u. 8, H-1081 Budapest, Hungary; frigyik@gmail.com

**Keywords:** forensic pathology, post mortem interval, multidisciplinary approach, machine learning, support vector machine

## Abstract

Henssge’s nomogram is a commonly used method to estimate the time of death. However, uncertainties arising from the graphical solution of the original mathematical formula affect the accuracy of the resulting time interval. Using existing machine learning techniques/tools such as support vector machines (SVMs) and decision trees, we present a more accurate and adaptive method for estimating the time of death compared to Henssge’s nomogram. Using the Python programming language, we built a synthetic data-driven model in which the majority of the selected tools can estimate the time of death with low error rates even despite having only 3000 training cases. An SVM with a radial basis function (RBF) kernel and AdaBoost+SVR provided the best results in estimating the time of death with the lowest error with an estimated time of death accuracy of approximately ±20 min or ±9.6 min, respectively, depending on the SVM parameters. The error in the predicted time ($t_{p}$[h]) was $t_{p}\pm 0.7$ h with a 94.45% confidence interval. Because training requires only a small quantity of data, our model can be easily customized to specific populations with varied anthropometric parameters or living in different climatic zones. The errors produced by the proposed method are a magnitude smaller than any previous result.

## 1. Introduction

The determination of the post mortem interval (PMI) is one of the oldest questions in forensic medicine, which has posed major challenges for experts since its inception and remains the focus of significant research. Both mathematical [1,2,3,4] and nonmathematical methods are used to address the problem [5]. The process of changing the temperature of the body involves a complicated interplay of various biological processes and factors, yet it is nevertheless characterized by physical laws. Early studies suggested that the Newtonian cooling law was unsuitable for mathematically characterizing the process because the cooling curve is sigmoidal rather than exponential due to a plateau phase [6,7,8]. The Marshall–Hoare formula, which was created empirically and contains a linear combination of two exponential functions [9,10,11], can be used to describe this sigmoidal curve. As it is a transcendental equation, if we want to determine the PMI, we can solve it numerically or graphically with the help of Henssge’s nomogram [12,13,14,15] or another simplified graphical solution [16]. Subsequent studies required the extension of the Marshall–Hoare formula with a weight-related correction factor [12,17,18,19,20], known as the Henssge formula, allowing for a more precise estimation of the PMI.

The mathematical description of the process has not changed since the introduction of the Henssge formula; however, multiple solutions for fitting empirical data using diverse methodologies have been developed: nonlinear least squares [21], conditional probability [22], Bayesian estimation [23,24], finite element simulation [25], Laplace transformation [26], numerical simulations [27,28,29], and neural networks [30]. A triple exponential model was another strategy to take into account; however, it did not yield the desired outcomes [31,32]. Brute-force calculations [33,34], heat-transfer modeling [25,35,36], the evaluation of a back-calculation [37], and a computational approximation in PHP [38] are other examples of unique approaches.

Machine learning is widely used in numerous fields of medical diagnostics and prognostics [39,40,41,42]. One possible method of estimating parameters is finding them by using the method of linear regression. We can choose from various mathematical tools to do so, such as different regression methods, decision trees [43], or applying different neural networks, in which the goal is to find an approximation to the model function belonging to a given learning set. A support vector machine (SVM) can be considered a special neural network, which is a supervised learning method that can have different kernel functions for its decision function [44,45,46]. The objective of the kernel method is to convert the original problem into a linearly solvable one. With its use, the data describing the problem to be solved are transformed into the kernel space through the application of nonlinear transformations, such as radial basis functions (RBF). The aim of our study was to analyze the accuracy of several different regression methods (decision tree [47], random forests [48], extra trees, bagging, AdaBoost, SVM, AdaBoost + SVM) and their combination in solving the aforementioned mathematical problem using the Python programming language.

The motivation behind this work comes from our desire to support the work of forensic experts by developing a modernized, flexible, and adaptive method that utilizes existing machine learning tools to enable a more accurate estimation of the PMI using present day training data than the commonly used Henssge nomogram and that can adapt to the constantly changing population.

## 2. Materials and Methods

The Henssge formula, and its graphical solution, the Henssge nomogram, are commonly used in methods to estimate the PMI:(1)$$\frac{T_{r}-T_{a}}{T_{0}-T_{a}}=A\cdot \exp\left(Bt\right)+\left(1-A\right)\cdot \exp\left(\frac{AB}{A-1}t\right),$$
where $T_{r}$ and $T_{a}$ are the rectal and environmental temperatures, respectively, measured at time *t*, $T_{0}=37.2$ °C is a constant representing the rectal temperature commonly assumed at the time of death. In the formula, *A* and *B* are parameters obtained empirically [17]. The value of the parameter A depends on the environmental temperature (Table 1).

The parameter *B* includes the body weight (*m*).
(2)$$B=-1.2815\cdot m^{-0.625}+0.0284$$

The Henssge formula in the two temperature ranges is as follows:

For $T_{a}\leq 23.2$ °C,
(3)$$\frac{T_{r}-T_{a}}{37.2-T_{a}}=1.25\cdot \exp\left(Bt\right)-0.25\cdot \exp\left(5Bt\right);$$
for $T_{a}\geq 23.3$ °C,
(4)$$\frac{T_{r}-T_{a}}{37.2-T_{a}}=1.11\cdot \exp\left(Bt\right)-0.11\cdot \exp\left(10Bt\right).$$

Any proper mathematical model should be capable of handling the uncertainties that can affect the accuracy of the resulting time of death, the estimate of which is based on the Henssge formula. As can be seen in Henssge’s nomogram, uncertainty can be caused by various factors, including the correction factor [49], body weight [12], and a variable environmental temperature and humidity [50]. From a practical point of view, a basic source of error may be the incorrect size ratio of the printed nomogram [51]. The Henssge nomogram graphically handles the uncertainties which can affect the accuracy of the determined PMI, while data-based models incorporate these uncertainties within themselves.

### 2.1. Data-Driven Model

The purpose of creating the data-driven model was to examine the estimation of the PMI using other mathematical methods. We decided to use decision trees (regression trees) and an SVM with an RBF kernel. Our model relied on the assumption that the generated data from which the system learned closely resembled reality. To create the model and perform the calculations, we used the Python programming language to generate data for learning and testing, which formed the basis of the theoretical model. We chose various regression trees and an SVM from the scikit-learn [52] package.

#### 2.1.1. The Generation of Data and Test Data

For training and testing the regression trees and the SVM, we used generated data. For each parameter that is required for the calculation using the Henssge formula, we randomly selected from a predetermined list of values. These were as follows:Time (h): 1–18, with a step of 0.5 h.Ambient temperature ( °C): −10–35, in increments of 0.5  °C.Correction factor: 0.7, 0.9, 1.0, 1.1, 1.2,1.3, 1.4, based on Table 5 in [49]Body weight (kg): between 50 and 100 kg, with a precision of 0.5 kg, drawn from a normal distribution with postselection (mean of 70 kg with σ large enough to generate an appropriate quantity of test data close to the upper limit).Rectal temperature ( °C). Based on the randomly selected data described above, it was calculated from the Henssge formula according to Algorithm 1 which uses Algorithm 2.The number of desired data points, which is an approximate value, since some of the weights drawn from a normal distribution were outside of the desired range and therefore were not considered in either the training or test data sets.

According to the literature [49], certain restrictions were taken into account when creating the data:(1)The ambient temperature must not be higher than the measured rectal temperature. Since the rectal temperature was calculated during data generation, this case could not occur.(2)For the correction factors, the value does not need to be adjusted based on weight until 1.4. Beyond this value, it must be corrected, but our model is currently not set up for this (see Table 5 in [49]).(3)In the case of the weight, the selection of the lower and upper limits was again based on the Table 5 in [49]. The selection of the 70 kg average was also based on Table 5 in [49].

Steps for generating the training and test data:(1)Randomly select one parameter set (weight, correction factor, environmental temperature) from the required sets of parameters for the Henssge formula.(2)Determine the rectal temperature by evaluating the Henssge formula.

In the pseudocode of Algorithm 3, the input parameter “count” is an integer that roughly determines the number of generated data points, as values outside the lower and upper bounds of the weight given by the normal distribution were also generated, but they were not used for training and testing.

The samples were separated into training and test data by dividing the total generated data set (X,y) into two parts. Usually, these parts are comprised of different percentages of data, with one part being used as the data to train the model, and the other part being used as the test data to evaluate the performance of the model. *X* is an n×4 dimensional matrix, where *n* is the number of actual data points that meet the conditions, and it contains the selected and calculated parameters in the following order $\left[m,cf,T_{r},T_{a}\right]$; *y* is an *n*-dimensional vector, where $y_{i},i=1\ldots n$ contains the randomly selected expected time interval for $X_{i}$. During the division, a prespecified percentage of the total data set was chosen for the test data and the rest was used for training.
**Algorithm 1:** Calculating rectal temperature.
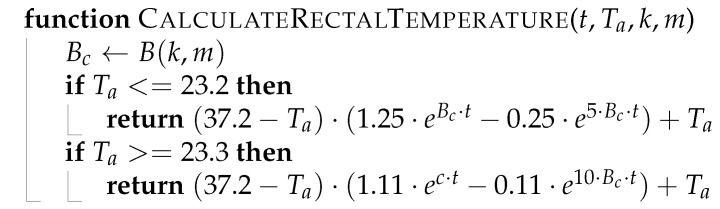

**Algorithm 2:** Body weight adjusted by correction factor.
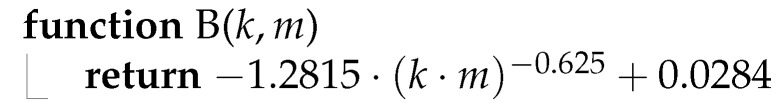

**Algorithm 3:** Generating training data and test data.
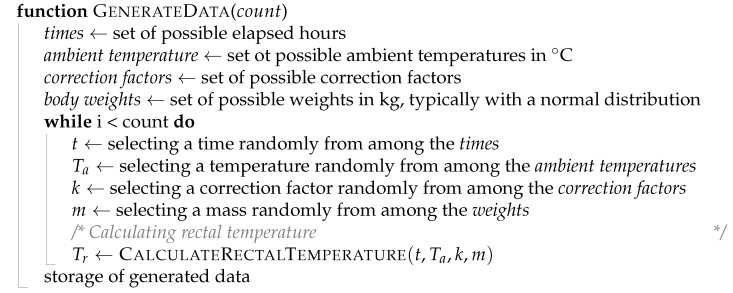



#### 2.1.2. Training

We began the process of training our selected regression models by utilizing the partial sample $\left(X_{train},y_{train}\right)$. This data set served as the input for our model training, which was performed using several different approaches.

One of the approaches we utilized was decision (regression) trees, including bagging, random forests, and extremely randomized trees. These techniques have been shown to be effective at modeling complex relationships between variables and making predictions in a variety of scenarios. In addition to the decision trees, we also used support vector regression (SVR) with a radial basis function (RBF) kernel. This is a powerful method that has been shown to be effective at modeling nonlinear relationships between variables. To further refine the results obtained from our extremely randomized trees and SVM, we applied a tree modified with an adaptive boosting method. This allowed us to improve the accuracy and precision of our model predictions.

Once our models had been successfully trained and optimized, we saved them for later use. This ensured that we could quickly and easily access our models and use them to make predictions in new scenarios, without having to repeat the time-consuming and computationally expensive training process.

#### 2.1.3. Testing

We tested the trained model with $\left(X_{test},y_{test}\right)$ subsets. The performance of the model was determined based on the mean absolute error (MAE), mean squared error (MSE), and coefficient of determination ($R^{2}$) values (see below). The results of the runs can be found in Appendix A Table A1–Table A6.

#### 2.1.4. Error Calculation

There are several mathematical tools available for determining the prediction error. Let *N* be the size of the sample and ${\hat{y}}_{i}={\hat{\beta }}_{0}+{\hat{\beta }}_{1}x_{i}$ the estimated value of $y_{i}$. The residual for the *i*th observation is defined as $e_{i}=y_{i}-{\hat{y}}_{i}$, that is, the difference between the expected value $y_{i}$ and the estimated value for the *i*th observation.

##### Sum of Squared Residuals (*SSR*)

In most cases, we minimized the sum of squared residuals (least-squares method).
(5)$$SSR=e_{1}^{2}+e_{2}^{2}+\cdots +e_{N}^{2}=\sum_{i=1}^{N}{\left(y_{i}-{\hat{y}}_{i}\right)}^{2}$$

##### Mean Squared Error (*MSE*)

The average of the squares of the differences between the estimated values and the actual values.
(6)$$MSE=\frac{\sum_{i=1}^{N}{\left(y_{i}-{\hat{y}}_{i}\right)}^{2}}{N}$$

##### Mean Absolute Error (*MAE*)

The average of the absolute differences between the estimated and actual values.
(7)$$MAE=\frac{\sum_{i=1}^{N}\left|y_{i}-{\hat{y}}_{i}\right|}{N}$$

##### Coefficient of Determination ($R^{2}$)

$R^{2}$ is a statistical measure that represents the proportion of the variance in the dependent variable that is predictable from the independent variable(s) in a regression model. $R^{2}$ ranges from 0 to 1, where 0 indicates that the model explains none of the variance in the dependent variable, and 1 indicates that the model explains all of the variance.
(8)$$R^{2}=\frac{TSS-SSR}{TSS},$$
where
(9)$$TSS=\sum_{i=1}^{N}{\left(y_{i}-\frac{1}{N}\sum_{i=1}^{N}y_{i}\right)}^{2}=\sum_{i=1}^{N}{\left(y_{i}-\bar{y}\right)}^{2}$$
is the sum of squared errors, where $\bar{y}$ is the mean value of the given data set.

## 3. Results

The accuracy of the mathematical model we used for estimating the PMI depended on the proper choice of the relatively large number of adjustable parameters. We considered a choice proper, if we obtained it through a learning process and the resulting model gave meaningful estimates in cases that were similar to those it had already encountered. If we tested a case that fell outside of the domain determined by the learned data, then the estimation error was supposed to grow. The version in this paper used the most commonly used correction factors from the set of all correction factors. The choice of environmental temperature range was based on the Henssge nomograms. One important factor influencing the quality of the generated data was the mean value (70 kg) and standard deviation (σ) of the normal distribution of the body weight, which determined the width of the Gaussian curve, which, in turn, set the range for the random weights obtained. When σ was very small, <5, the generated weights were in a very small range with a very high probability, but if σ was chosen large enough, the data were selected from a larger set. The goal was to have enough data for training with weights between 50 and 100 kg. We determined the σ through multiple trials and for ≈11,000 generated data, and we found that σ=10 was already sufficient.

The data were generated such that for each body weight, we were randomly selecting $T_{a}$, the correction factor, and the expected time of death using a uniform distribution. Then, $T_{r}$ was calculated based on these values.

In order to examine the results of the theoretical model, we trained and tested the model using a variety of number of cases and methods, so as to find the tool working with the smallest error for solving the problem.

We designated 25% of the generated data as test data and utilized the remainder for training the system. As the foundation for the theoretical model, we employed various regression tools with differing configurations and sought out the best parameterization for each tool individually. Following this, by using the mathematical tools in combination, we further improved the results obtained. The methods investigated were as follows:Regression tree;Random forests;Extremely randomized trees;Tree modified with the bagging method;SVR with an RBF kernel;SVR improved with adaptive boosting.

### Results of Training

The training and estimation time, the errors (*MAE*, *MSE*, $R^{2}$), and the best parameterization of the regression tools tested with various parameterizations are presented in the tables in Appendix A Table A1–Table A6 for each method. The number of generated data was increased by a thousand, minus the number of cases that did not fall within the determined range of 50–100 kg.

Based on the results, it can be concluded that with a larger training data set, all methods were capable of estimating the time of death with a decreasing error, as shown in the graphs in Figure 1. According to both the MAE, MSE, and the $R^{2}$ (≈1) values, the best result was achieved by the combined use of SVR and an adaptive regression tree, as this method further improved the results obtained by SVR [53,54]. Since in the used Python implementation, the C parameter represents the compromise between minimizing false classification errors and maximizing the decision boundary, meaning the higher the value of C, the fewer the false classifications and the stricter the decision margin, we performed four additional control runs with higher C values (10, 20, 50, 100) to check the accuracy of the SVR estimate when improved by adaptive boosting for these four cases as well. The results of these were divided into two parts, first for SVR alone, and then for the improved results using the adaptive boosting method.

According to Figure 1, it can be observed that most selected mathematical tools were able to estimate the time of death with low error rates even with a minimum of 3000 training examples, based on the current settings. However, the decision tree was an exception, as it still produced high errors compared to the others, even with over 10,000 data points.

The results obtained with SVR and AdaBoost + SVR models using the parameters C = 50 and C = 100 at at a sample size of approximately 11,000 were as follows: Based on Table 2 and Table 3, it can be concluded that increasing the value of C further improved the achieved results. By breaking down the average error of the 25% of test data into correction factors with a 5 kg binning, we determined for the cases of C = 5 and C = 100 (see in Figure 2) that in the former case, the error was approximately ±0.3 h = 20 min, and in the latter case, the two worst results were approximately ±0.16 h = ±9.6 min, but the average errors were below 4 min.

After comparing the results of various selected methods, it can be concluded that the two best results were obtained with SVR and AdaBoost + SVR, as can be seen in Figure 3 and Table 4. This is due to the fact that these two methods had the most test results within 1σ of the mean.

## 4. Discussion

The Henssge formula and its graphical solution, the Henssge nomogram, are commonly used to estimate the PMI. However, uncertainties—including the correction factor, body weight, and variable environmental conditions—can affect the accuracy of the resulting time interval. The Henssge nomogram handles these uncertainties graphically, while our model incorporated these uncertainties within itself. In other words, our model did not require the use or knowledge of the Marshall–Hoare or Henssge formula with correction factors, and it did not contain any empirical variables. The generated data closely resembled the reality and formed the basis of our theoretical model. Our data-driven model showed that an SVM with an RBF kernel and the AdaBoost + SVR method provided the best results in estimating the time of death with the lowest error. The estimated accuracy of the time of death was approximately within ±20 min or ±9.6 min, depending on the SVM parameters used. The predicted time error was $t_{p}\pm 0.7$ h with a 94.45% confidence interval. When compared to the Henssge nomogram, where the accuracy was claimed to be ±2.8 h for both temperature ranges when correction factors were applied, it can be concluded that the created model was capable of estimating the time of death with a sufficient accuracy while taking into account the constraints based on the learned data set. The significant differences and errors arose from the fact that the model encountered these cases with fewer samples during the learning process, but they still fell within the accuracy zone determined by the nomogram. The current limitations of the theoretical model are the number of correction factors, a maximum time interval of 18 h, the need for the training data to be provided with an accuracy of 30 min, and a body weight limited to 50–100 kg. Based on the results presented in Figure 1, it can be inferred that most of the mathematical tools used in this study were able to accurately estimate the time of death with relatively low error rates, even with a minimum of 3000 training examples under the current settings. One notable advantage of our models was that they required very few data for training, which means that they can be applied in various geographical regions, including smaller areas. This feature makes the model highly versatile and adaptable to specific populations with differing anthropometric characteristics or living in different climate zones, because it can be trained with real, available data. Moreover, the model can be easily adapted to suit one’s needs, making it an ideal tool for a range of settings and situations.

Most articles on this topic determine the PMI using basic physics or numerical calculations. However, these results cannot be compared to ours because we used the Henssge formula to generate synthetic data. To the best of our knowledge, there is only one paper that used neural networks (multilayer feedforward networks) to address this problem [30]. Zerdazi and coworkers constructed a network using MATLAB 2012 with two layers. The first layer, called the hidden layer, contained 10 neurons, each using the hyperbolic tangent as an activation function. The second layer, known as the output layer, had only one neuron, which employed a linear activation function.

Our method achieved much better results with 257 cases than Henssge’s solution. While our model used a different machine learning approach, we can compare our results to theirs (see Table 5 and Table 6). To obtain the best comparability, we trained our model again with the same features as those described in that paper [30]. Two scenarios were investigated with common features: an environmental temperature ranging from 4.5  °C to 18  °C, 20% of data for validation, and 20% for testing.

Scenario 1: 275 observations, time of death between 20 min and 18 h. The results obtained are as follows:

**Table 5 diagnostics-13-01260-t005:** Comparison of our SVM and AdaBoost + SVM results with the result from the multilayer feedforward network (neural method) by Zerdazi et al. [30] in case of Scenario 1.

Name	MAE	MSE
Neural method	1.85	5.69
SVR	0.17	0.14
AdaBoost + SVR	0.17	0.12

Scenario 2: 184 observations, time of death less than 7 h. The results obtained are as follows:

**Table 6 diagnostics-13-01260-t006:** Comparison of our SVM and AdaBoost + SVM results with the result of the multilayer feedforward network (neural method) by Zerdazi et al. [30] in case of Scenario 2.

Name	MAE	MSE
Neural method	0.86	1.21
SVR	0.18	0.08
AdaBoost + SVR	0.14	0.05

From these results, we can conclude that the proposed method performed much better: all the errors were at least one order of magnitude smaller than the results in [30].

Further testing with real data is needed. The future goal is to develop a phone or web application based on the model with a graphical interface for easier use and to create a database for storing anonymized training data.

## 5. Conclusions

Our research demonstrated that the estimated PMIs produced by our models using existing machine learning tools such as SVMs and decision trees were far more satisfactory than those produced by the Henssge formula or the method utilizing neural networks. In contrast to traditional mathematical methods, including the Henssge nomogram, that yield fixed formulas and whose performance remains constant, our models can be continuously improved because training can be resumed whenever new additional data are available. As our models estimated the PMI with low error rates even with only 3000 training cases, they can be easily adapted to specific populations with different characteristics or living in different climatic zones.

## Figures and Tables

**Figure 1 diagnostics-13-01260-f001:**
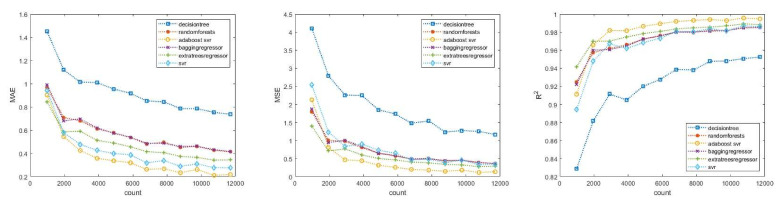
MAE, MSE, and $R^{2}$ of theoretical model.

**Figure 2 diagnostics-13-01260-f002:**
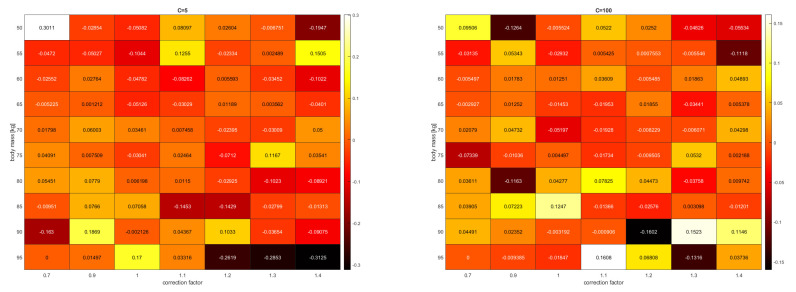
Average error with a 5 kg windowing as a function of the correction factor for C = 5 and C = 100 cases with the AdaBoost + SVR model.

**Figure 3 diagnostics-13-01260-f003:**
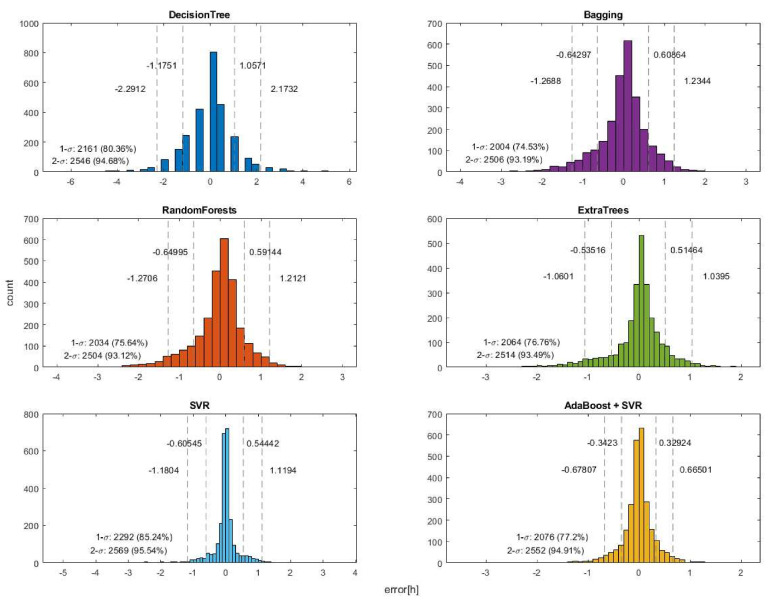
The results of the different methods at distances of 1σ and 2σ.

**Table 1 diagnostics-13-01260-t001:** The value of the parameter *A*.

$T_{a}$	*A*
≤23.2 °C	1.25
≥23.3 °C	1.11

**Table 2 diagnostics-13-01260-t002:** The errors of SVR.

	MAE	MSE	$R^{2}$
C = 10	0.2578	0.2746	0.9886
C = 20	0.2255	0.2252	0.9906
C = 50	0.1979	0.1828	0.9924
C = 100	0.1683	0.1290	0.9947

**Table 3 diagnostics-13-01260-t003:** The errors of AdaBoost + SVR.

	MAE	MSE	$R^{2}$
C = 10	0.2177	0.1340	0.9944
C = 20	0.1875	0.0987	0.9959
C = 50	0.1820	0.1109	0.9954
C = 100	0.1606	0.0762	0.9969

**Table 4 diagnostics-13-01260-t004:** The results of the different methods at distances of 1σ and 2σ.

Name	1σ Value	2σ Value	1σ	2σ
Decision tree	−1.1751–1.0571	−2.2912–2.1732	2161 (80.36%)	2546 (94.68%)
Bagging	−0.64297–0.60864	−1.2688–1.2344	2004 (74.53%)	2506 (93.19%)
Random forests	−0.64995–0.59144	−1.2706–1.2121	2034 (75.64%)	2504 (93.12%)
Extra trees	−0.5316–0.51464	−1.0601–1.0395	2064 (76.76%)	2514 (93.49%)
SVR	−0.60545–0.54442	−1.1804–1.1194	2292 (85.24%)	2569 (95.54%)
AdaBoost + SVR	−0.3423–0.32924	−0.67807–0.66501	2076 (77.2%)	2552 (94.91%)

## Data Availability

The source code of our model is available: https://github.com/livdan/TOD (accessed on 6 March 2023).

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
