# Peer review of "Beyond Henssge’s Formula: Using Regression Trees and a Support Vector Machine for Time of Death Estimation in Forensic Medicine"

_diagnostics, 2023, doi:10.3390/diagnostics13071260_

Round 1

Reviewer 1 Report

The work is somehow new but several limitations from the English to the technical hinder to grasp the main point of the work, some of them are listed below:

My Comments and Suggestions to Authors:

1- In my opinion, the abstract is too cumbersome and is hard to catch the key point. The keywords need to be more detailed.

2- There are many typos and grammatical errors in the manuscript. It is strongly suggested that the whole work to be carefully checked by someone who has expertise in technical English writing.

3- Manuscript organization needs to be restructured. Kindly be as concise and straight to the point as possible.

4- In the Introduction section, the new features of the proposed method and the main advantages of the results over others should be clearly described.

5- The contributions presented in this paper are not sufficient for possible publication in this journal. I highly suggest authors to clearly define the contributions.

6- The problems of this work are not clearly stated. There is ambiguity in statement understanding.

7- The motivation is not clear to me. The authors need to explain why we need this work and how it is different from the state-of-art schemes and present more details about the novelty.

8- The conclusions in this manuscript are primitive. Write your conclusions.

Author Response

We appreciate the time and effort that you have dedicated to providing your valuable feedback on our manuscript. We have been able to incorporate changes to reflect most of the suggestions provided by you. Here is a point-by-point response to your comments and concerns:

Ad 1

In my opinion, the abstract is too cumbersome and is hard to catch the key point. The keywords need to be more detailed.

As suggested by the Reviewer, we totally revised and expanded the abstract. We also updated the keywords: ‘forensic pathology; post mortem interval; multidisciplinary approach; machine learning; support vector machine’

Ad 2

There are many typos and grammatical errors in the manuscript. It is strongly suggested that the whole work to be carefully checked by someone who has expertise in technical English writing.

Our manuscript has been reviewed by a colleague and revised to improve readability.

Ad 3

Manuscript organization needs to be restructured. Kindly be as concise and straight to the point as possible.

We would like to thank for the constructive suggestions. Many sentences of the manuscript have been carefully rewritten or reorganized to enhance the logic flow. We also extended the ‘Introduction’ and the ‘Discussion’ sections as well as the ‘Conclusions’ section. According to these changes, we added new references. We truly hope that the revised manuscript is clear enough to follow.

Ad 4

In the Introduction section, the new features of the proposed method and the main advantages of the results over others should be clearly described.

We extended the ‘Introduction’ section and we added a sentence at the end of the section to show our motivation for this work. The main advantages of our method and our results became parts of the ‘Discussion’ section to enhance the logic flow. We hope the Reviewer will find these changes satisfactory.

Ad 5

The contributions presented in this paper are not sufficient for possible publication in this journal. I highly suggest authors to clearly define the contributions.

We agree with the reviewer that our contribution was not clearly stated. So, we have made it explicit in the ‘Discussion’ section and throughout the whole paper.

Ad 6

The problems of this work are not clearly stated. There is ambiguity in statement understanding.

We extended the ‘Introduction’ and the ‘Discussion’ sections as well as the ‘Conclusions’ section with a hopefully much clearer problems statement and discussion.

Ad 7

The motivation is not clear to me. The authors need to explain why we need this work and how it is different from the state-of-art schemes and present more details about the novelty.

We fully agree with the reviewer. Our extended ‘Introduction’ contains more detailed information about the current state-of-art in forensic medicine and about our method in other fields. On the other hand, a large chunk of the ‘Discussion’ is dedicated to the comparison of our results to the best results that we could find in the literature. We found one paper (see reference 30) that uses neural networks (Multilayer Feedforward Networks) to address the problem of PMI estimation. To obtain the best comparability, we trained our model again with the same features as those described in this new reference paper. From the results, we can conclude that our method performs much better, because all the errors were at least one order of magnitude smaller than their results (see in Table 5 and Table 6). Our research demonstrates that the estimated PMIs produced by our models using existing machine learning tools such as SVM and decision trees are far more satisfactory than those produced by the Henssge formula or the method utilizing neural networks. We highlighted the advantages of our method over the existing ones.

Ad 8

The conclusions in this manuscript are primitive. Write your conclusions.

As suggested by the reviewer, we rewrote the ‘Conclusions’ section to highlight the novelty and future practical potential of our work. 

Reviewer 2 Report

1)    The technique of this paper is well-known. The authors must clearly show the difference and improvements in comparison with the existing results in the view of technique analysis. 2)    The motivation on why to propose such a framework and strategy in real-world applications should be clearly emphasized. It would be much better if some guideline remark words on practical applications should be given. 3)    Explain the feasibility of the results from the implementation and computational point of view. Some remarks on computation complexity of the results should be given. 4)    Update the recent references related to this work

Author Response

We appreciate the time and effort that you have dedicated to providing your valuable feedback on our manuscript. We have been able to incorporate changes to reflect most of the suggestions provided by you. Here is a point-by-point response to your comments and concerns:

Ad 1

The technique of this paper is well-known. The authors must clearly show the difference and improvements in comparison with the existing results in the view of technique analysis.

Indeed, the techniques we were used are very well-known. The aim of the paper was to report on applying these well-known techniques to the problem of determining the post mortem interval. Our extended ‘Introduction’ contains more detailed information about the current state-of-art in forensic medicine and about our method in other fields. On the other hand, a large chunk of the ‘Discussion’ is dedicated to the comparison of our results to the best results that we could find in the literature. Our research demonstrates that the estimated PMIs produced by our models using existing machine learning tools such as SVM and decision trees are far more satisfactory than those produced by the Henssge formula or the method utilizing neural networks. We hope the Reviewer will find these changes satisfactory.

Ad 2

The motivation on why to propose such a framework and strategy in real-world applications should be clearly emphasized. It would be much better if some guideline remark words on practical applications should be given.

We would like to thank for the constructive suggestions. The motivation, which we made explicit in the introduction, comes from a real-world problem, namely the problem of determining the post mortem interval. What we are planning to do is to develop an application that supports professionals who want to use our method. Many sentences of the manuscript have been carefully rewritten or reorganized to enhance the logic flow. We also extended the ‘Introduction’ and the ‘Discussion’ sections as well as the ‘Conclusions’ section. According to these changes, we added new references.

Ad 3

Explain the feasibility of the results from the implementation and computational point of view. Some remarks on computation complexity of the results should be given.

Feasibility was never really a question, but efficiency was. Everything was implemented using standard computational functions, whose computational complexity is well-known and the analysis of them was not a goal of this study. We focused on the future application. According to Reviewers suggestion, we extended the ‘Introduction’ section and the ‘Discussion’ and ‘Conclusions’ sections to highlight the novelty and future practical potential of our work.

Ad 4

Update the recent references related to this work.

We have extended the list of references with many contemporary works/papers. Furthermore, we found one paper (see reference 30) that uses neural networks (Multilayer Feedforward Networks) to address this problem. To obtain the best comparability, we trained our model again with the same features as those described in this new paper. From the results, we can conclude that our method performs much better, because all the errors were at least one order of magnitude smaller than their results (see in Table 5 and Table 6).

Round 2

Reviewer 1 Report

In. the revised manuscript, the authors have addressed all my concerns.  Overall, this work is in good quality and have met the requirement of Diagnostics. I suggest to accept this manuscript.